# Video Timeline Modeling For News Story Understanding

**Meng Liu**[1], **Mingda Zhang**[2], **Jialu Liu**[2], **Hanjun Dai**[2], **Ming-Hsuan Yang**[2], **Shuiwang Ji**[1], **Zheyun Feng**[2], **Boqing Gong**[2]

[1]Texas A&M University
[2]Google
{mengliu,sji}@tamu.edu,
{mingdaz,jialu,hadai,minghsuan,zfeng,bgong}@google.com

## Abstract

In this paper, we present a novel problem, namely video timeline modeling. Our objective is to create a video-associated timeline from a set of videos related to a specific topic, thereby facilitating the content and structure understanding of the story being told. This problem has significant potential in various real-world applications, for instance, news story summarization. To bootstrap research in this area, we curate a realistic benchmark dataset, YouTube-News-Timeline, consisting of over 12k timelines and 300k YouTube news videos. Additionally, we propose a set of quantitative metrics to comprehensively evaluate and compare methodologies. With such a testbed, we further develop and benchmark several deep learning approaches to tackling this problem. We anticipate that this exploratory work will pave the way for further research in video timeline modeling. The assets are available via `https://github.com/google-research/google-research/tree/master/video_timeline_modeling`.

## 1   Introduction

As the amount of online video content continues to grow rapidly, organizing and browsing through this massive collection of videos become increasingly challenging, especially for news videos. In the modern world, the news is not just delivered through text and images, but also through videos. With the increasing use of video streaming platforms and emerging short video applications, the number of news videos is growing at an unprecedented pace. However, such a vast number of news videos are usually unstructured. For example, there are tens of thousands of news videos about the topic "COVID-19 Developments in 2020" on YouTube, and their relationships are not well presented. It would be nearly impossible for users to watch such numerous unstructured videos, thereby hardly obtaining an in-depth understanding of the news story.

In this work, we propose a video timeline modeling problem to help analyze and comprehend news videos. In particular, as illustrated in Figure 1, our goal in this problem is to construct a video-associated timeline that represents the critical events and their evolutionary order in a news story. This is of great importance for a better, user-friendly understanding of news stories. To the best of our knowledge, this is the first work to define and formulate this problem.

To facilitate research into this problem, we present a benchmark dataset, namely YouTube-News-Timeline, by crawling timelines from the web and sourcing videos from the YouTube news video corpus. YouTube-News-Timeline has over 12k timelines and covers over 300k YouTube news videos in total. To quantitatively evaluate methods on the proposed dataset, we further design several metrics to assess the complex prediction results. Aside from our contributions of the standard benchmark, we

37th Conference on Neural Information Processing Systems (NeurIPS 2023) Track on Datasets and Benchmarks.

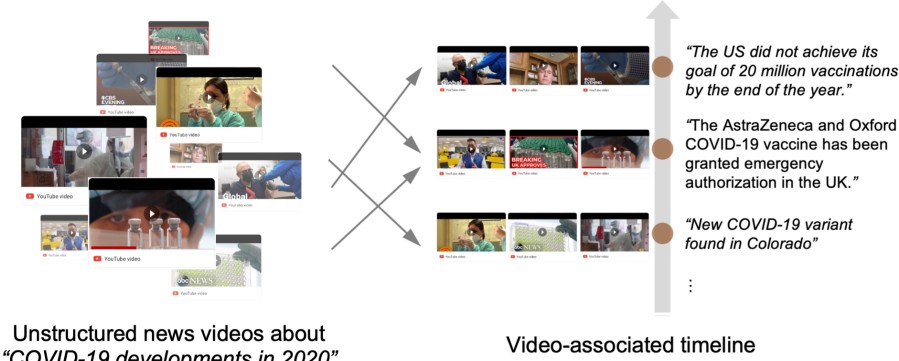

Figure 1: Illustration of video timeline modeling.

further propose several novel deep learning approaches, based on Transformer [Vaswani et al., 2017], pointer networks [Vinyals et al., 2015], and knowledge distillation [Hinton et al., 2015], to tackle the problem. It is demonstrated through extensive experiments that our preliminary methods are effective and could serve as reference baselines for future research.

## 2  Related Work

Timeline modeling for textual data is a well-studied problem in the field of natural language processing [Ghalandari and Ifrim, 2020]. The goal is to identify key events and provide short summaries from a collection of news articles or other text corpora. Existing work either extracts important sentences [Allan et al., 2001, Chieu and Lee, 2004, Radev et al., 2005, Yan et al., 2011, Nguyen et al., 2014, Tran et al., 2015, Martschat and Markert, 2018, Camacho Barranco et al., 2019] or generates abstractive summaries [Steen and Markert, 2019] to construct the desired timeline. These methods solely focus on using textual data and ignore visual signals which contain rich context that may help enhance the understanding of news stories. Wang et al. [2016] jointly learn from text and image data to generate timelines. Our work moves one important step forward by modeling timelines for videos.

While general video classification and representation tasks [Ji et al., 2012, Misra et al., 2016, Fernando et al., 2017, Han et al., 2019, Piergiovanni et al., 2020] have established importance in the field, they focus on understanding individual videos. In contrast, the video timeline modeling task considers the intricate relationships among multiple videos. We also note that the video timeline modeling problem is distinct from video summarization [Ngo et al., 2003, Khosla et al., 2013, Gong et al., 2014, Zhang et al., 2016, Li et al., 2018, Huang and Wang, 2019, Apostolidis et al., 2021]. Existing video summarization works mainly create a concise outline by selecting informative parts from a single video, while our task involves modeling the complex relationships among multiple videos. In addition, while video summarization typically relies on low-level visual features like appearance and motion, our video timeline modeling task requires a higher-level understanding of the events contained in the videos and the ability to align events across multiple videos.

Another relevant task is multi-video summarization [Panda et al., 2017, Wu et al., 2020], which primarily aims to create a concise representation of multiple videos by selecting keyframes or short segments, emphasizing the overlap and complementarity among the videos. On the other hand, while video timeline modeling also takes multiple videos as input, it seeks to identify, order, and represent significant events in an evolutionary sequence. Our task is also relevant to the recently proposed visual abductive reasoning (VAR) task [Liang et al., 2022, Hessel et al., 2022], which aims to infer the hypothesis that can best explain the premise, given an incomplete set of multiple visual events. Both VAR and our task involve multiple visual inputs, but there are significant differences. VAR reasons the causes of observed events, whereas we infer the evolutionary order of events within a set of videos. In addition, while VAR typically involves a single visual event, video timeline modeling deals with multiple videos that may contain various scenes and events.

# 3 Proposed Benchmark

## 3.1 Video Timeline Modeling Problem

Given a set of videos belonging to a specific news topic, the goal of the video timeline modeling problem is to yield a number of **ordered** nodes to form a timeline, where each node corresponds to a set of videos selected from the input video set. Formally, each training timeline sample consists of a video set $V = \{v_1, v_2, \ldots, v_N\}$ and their corresponding labeled node IDs $y = \{a_1, a_2, \ldots, a_N\}$, where $a_i \in \{1, 2, \ldots, K\}$ denotes the labeled node ID of the $i$-th video and $K$ is the number of nodes on the timeline. The objective of video timeline modeling is to learn a function mapping: $f : \{v_1, \ldots, v_N\} \mapsto \{a_1, \ldots, a_N\}$. The $k$-th node can be represented as a set $C_k = \{v_i | a_i = k, \forall i\}$. For a concise problem definition, we assume that each video must be assigned to and can only be assigned to one node, $i.e.$, $\bigcup_{k=1}^{K} C_k = V$ and $\forall k \neq l, C_k \cap C_l = \emptyset$.

As shown in Figure 1, intuitively, the objective of the proposed problem is to arrange a collection of unstructured news videos related to a specific news topic into a coherent timeline story. Particularly, each node on the obtained timeline represents a key event of the news topic, and the order among nodes denotes the evolution of the news story. Note that this work, as an exploratory attempt, focuses solely on assigning input video sets to ordered nodes. Once the ordered nodes are obtained, generating event information (in text, key frames, shorter videos, $etc.$) through video summarization becomes straightforward. To avoid any confusion, we consider video summarization as a separate step and do not include it in the problem we have defined.

Note that, in our problem setting, the textual timeline including the desired number of nodes is not given as an input. This setup aligns with the realistic scenario where a well-organized and timely timeline is typically unavailable for news, particularly for breaking news, but we are able to accurately retrieve the videos related to a news topic based on keyword filters or embedding guided retrieval [Revaud et al., 2013, Kordopatis-Zilos et al., 2019]. Another setting, where an authoritative timeline is provided as input and video assignments (or, recommendations) are conditional on that timeline, might also be practically useful. Both setups are valid, each stemming from distinct problem assumptions and targeting different real-world situations.

## 3.2 YouTube-News-Timeline Dataset

**Challenges.** Although the news video timeline modeling problem has significant potential in real-world applications, this area is unexplored. The main reason is likely that it is challenging to obtain such a video-associated timeline dataset. We summarize the main challenges as follows. i) News stories are often complex and multifaceted, with multiple events happening simultaneously or in rapid succession. As a result, creating a comprehensive and accurate timeline can be time-consuming and require a high level of expertise in the subject matter. ii) There does not exist a golden timeline for a news topic, since such a timeline is highly subjective, with different resolutions on the time scale and the event scale. iii) Assuming we have an authoritative reference timeline, it is still nearly impossible to use human labeling service to watch such an overwhelming number of videos and assign them to nodes on the reference timeline.

**Data collection pipeline.** Given such challenges, we consider using an automatic pipeline, instead of human labeling, to collect video-associated timeline samples. Our overall pipeline is illustrated in Figure 2. We start by crawling timelines from articles on the web that have mentioned the "timeline" keyword in the title or URL and have ordered dates in the content. We reasonably assume the crawled timelines can be considered authoritative and comprehensive for two reasons. i) Such articles are typically written by journalists or experts in their respective fields, who have access to reliable sources of information and are trained to ensure that the information presented is factual and of high quality. ii) The timelines themselves are presented in a structured and organized manner, with clear dates and descriptions of the events that occurred, thereby making them convincing and easy to be verified.

Using HTML parsing, for each crawled timeline, we then extract the chronological dates for the key events and their corresponding textual descriptions, specifically scraping the text descriptions from the corresponding leading paragraph. If the leading paragraph length is below 80 characters, we supplement it by appending additional words from subsequent paragraphs. This ensures we obtain a robust volume of information for the ordered nodes on the timeline. Next, we employ an embedding-based retrieval technique to identify relevant videos, from the in-house YouTube news

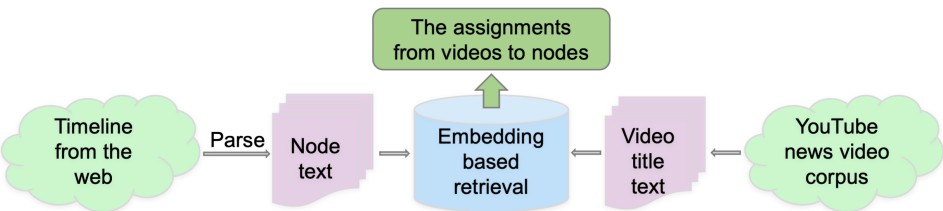

Figure 2: Illustration of our data collection pipeline.

video corpus, that correspond to the textual descriptions of nodes on the timeline. For both node text and video title text, we use the embedding given by the pre-trained NewsEmbed model [Liu et al., 2021], a universal text encoder for the news domain. By comparing the cosine similarity between these embeddings, we retrieve appropriate videos for each node text description on the timeline. This approach enables us to efficiently and effectively curate a collection of news videos that accurately reflect the context of the timeline nodes. Such assignments from videos to nodes are the desired labels for the video timeline modeling problem. Figure 3 illustrates an example of a crawled timeline and the corresponding video-associated timeline obtained via the above process.

**Sanity assurance.** To ensure that the retrieved videos are correct in terms of timeline node assignment, we prioritize high precision over high recall in the embedding-based retrieval process by setting a high embedding similarity threshold. In addition, we retain a maximum of the top 5 videos that surpass this stringent criterion, ensuring the fidelity of the node-video assignment. Furthermore, we take into account the possibility of consecutive events being semantically similar, which can result in similar embeddings, thus retrieving common videos. To mitigate this, we filter the videos associated with each node to only retain those with a release time, *i.e.*, the time when the video is published on

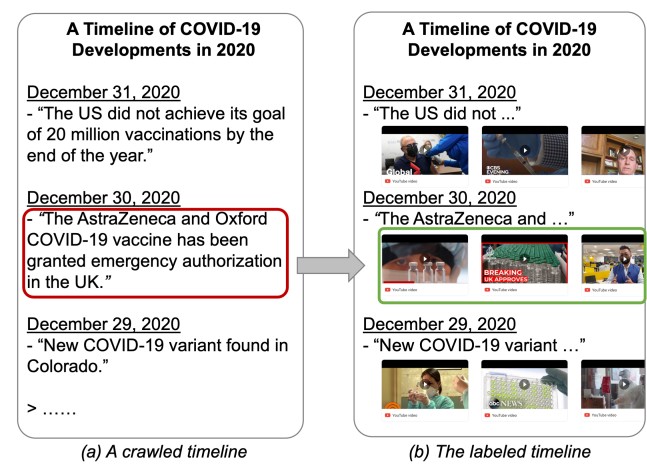

(a) A crawled timeline    (b) The labeled timeline

Figure 3: Example of (a) a crawled timeline and (b) the obtained video-associated timeline.

YouTube, falling between the two consecutive timestamps indicated by the current node and the following node. This approach helps reduce any ambiguity between consecutive events and ensures a higher level of precision in the assigned videos. According to our manual check for sampled timelines, the above strategies are effective to ensure the collected labels are of high quality.

**Dataset overview.** For each timeline sample, we provide the following information. (1) The URL link of the webpage where we crawl the timeline. (2) The URL links of the retrieved

| #Timelines | #Nodes | #Videos |
|---|---|---|
| 9936/1255/1220 | 74886/9325/9171 | 242685/30369/29930 |

Table 1: Statistics of the training/validation/testing subsets.

YouTube news videos that are related to the specific news topic. (3) The corresponding assignment from videos to nodes. Such assignments are obtained by the described data collection pipeline. A data format example is included in the appendix.

Note that in the YouTube-News-Timeline dataset, we provide the Youtube URL link for each video. Thus, any public metadata that can be crawled from YouTube can be used as input features. In Section 4, we use the same set of input features when comparing different methods.

**Dataset statistics.** We crawl over 20k timelines and collect over 12k qualified video-associated timelines after filtering out certain failure cases during the data collection process. These timelines include over 300k YouTube news videos in total. The duration of these videos ranges from 3 seconds to 12 hours, and their average duration is around 10 minutes. We randomly split the timelines into

training, validation, and testing subsets with a ratio $0.8/0.1/0.1$ before the data collection process. The number of timelines, nodes, and videos on each split in the final dataset are summarized in Table 1. Additionally, we present important distributions for each split in Figure 4, including the number of videos per node, the number of nodes per timeline, and the number of videos per timeline. Note that the number of nodes on a timeline in our dataset is between 2 and 24. Further details on the dataset's characteristics, such as news sources, topics, and chronological information about events, can be found in the appendix. Importantly, our reference timelines are sourced from a diverse set of publishers, cover a wide array of topics, and span an expansive temporal range, ensuring diversity and alleviating inherent biases.

## 3.3 Evaluation Protocol

To quantitatively evaluate methods on the proposed dataset, we further develop our evaluation protocol to cover several metrics. Following our previous notations, for a specific news topic, we have a set of input videos, denoted as $V = \{v_1, v_2, \ldots, v_N\}$, and their corresponding labeled node IDs are $y = \{a_1, a_2, \ldots, a_N\}$. We further denote the predicted node IDs as $\hat{y} = \{\hat{a}_1, \hat{a}_2, \ldots, \hat{a}_N\}$, where $\hat{a}_i \in \{1, 2, \ldots, \hat{K}\}$ represents the predicted node ID for video $v_i$. Here, $\hat{K}$ is the number of nodes produced by the prediction, which may be different from $K$ since the desirable number nodes is not known as an input in our problem setting. Accordingly, let

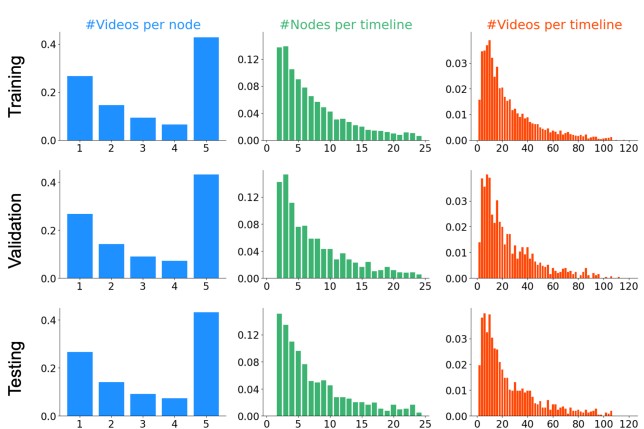

Figure 4: Distributions of the number of videos per node, the number of nodes per timeline, and the number of videos per timeline in the training, validation, and testing subsets.

$C_k \subseteq V$ for $k \in \{1, 2, \ldots, K\}$ and $\hat{C}_l \subseteq V$ for $l \in \{1, 2, \ldots, \hat{K}\}$ denotes the $k$-th node on the labeled timeline and the $l$-th node on the predicted timeline, respectively. Note that the number of input videos $N$ and the desired number of nodes on the timeline $K$ vary across samples, as depicted in Figure 4. For quantitative evaluation, we propose to measure two essential aspects according to the problem characteristics. In particular, we measure *the correct identification of events on the timeline* and *the correct evolution ordering of videos on the timeline*. We describe the proposed metrics in detail below.

**Example.** To enhance understanding, we use the following dummy example to help explain these metrics. In the example, we have 10 input videos belonging to a news topic, denoted as $\{v_1, v_2, \ldots, v_{10}\}$. The labeled node IDs are $\{a_1, a_2, \ldots, a_{10}\} = \{1, 1, 1, 1, 1, 2, 2, 2, 2, 3\}$. Suppose the predicted node IDs as $\{\hat{a}_1, \hat{a}_2, \ldots, \hat{a}_{10}\} = \{1, 1, 1, 1, 3, 4, 2, 2, 4, 3\}$. Obviously, there are 3 nodes on the labeled timeline, including $C_1 = \{v_1, v_2, v_3, v_4, v_5\}$, $C_2 = \{v_6, v_7, v_8, v_9\}$, and $C_3 = \{v_{10}\}$, while we have 4 nodes on the predicted timeline, including $\hat{C}_1 = \{v_1, v_2, v_3, v_4\}$, $\hat{C}_2 = \{v_7, v_8\}$, $\hat{C}_3 = \{v_5, v_{10}\}$, and $\hat{C}_4 = \{v_6, v_9\}$.

**Metric 1: Node-level precision and recall.** To measure how well the events on the constructed timeline match the ground truth events, we propose to compute precision and recall for nodes. First, we calculate the IoU scores between the ground truth nodes and the predicted nodes. Mathematically, the IoU score between node $C_k$ and node $\hat{C}_l$ is computed as $\frac{|C_k \cap \hat{C}_l|}{|C_k \cup \hat{C}_l|}$. Next, we construct a bipartite graph with the ground truth nodes and the predicted nodes as two separate sets of vertices. Each vertex is

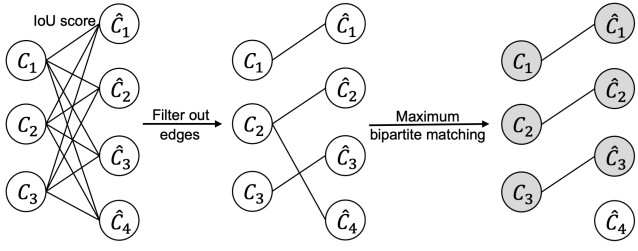

Figure 5: Illustration of matching ground truth nodes and predicted nodes.

connected to every vertex from the opposite set with edge weights equal to the corresponding IoU score. We further consider an edge to be a valid match only if its weight is greater than or equal to a threshold $\sigma$. All other edges are removed from the bipartite graph. Then, our goal is to find the maximum number of predicted nodes that match ground truth nodes, which is equivalent to a maximum bipartite matching problem, where the goal is to select a maximum number of edges in a bipartite graph in such a way that no two edges share a vertex. Such a problem can be converted to a maximum flow problem and solved via the Ford–Fulkerson algorithm [Ford and Fulkerson, 1956]. With the number of correctly matched nodes, we can compute the desired precision and recall as follows. The precision is the ratio of the number of correctly matched predicted nodes to the total number of predicted nodes, while the recall is the ratio of the number of correctly matched ground truth nodes to the total number of ground truth nodes. In our experiments, we select $\sigma = 0.5$. The node matching for our example is shown in Figure 5. The found matched node pairs are $(C_1, \hat{C}_1)$, $(C_2, \hat{C}_2)$, and $(C_3, \hat{C}_3)$. Thus, the precision and recall for this timeline sample are $\frac{3}{4}$ and $1$, respectively.

**Metric 2: Video-level Hamming distance.** In order to evaluate the correctness of video ordering, we propose to calculate the Hamming distance between the predicted node IDs and ground truth node IDs. Specifically, Hamming distance between two sequences is the number of positions at which the corresponding symbols are different. This can directly reflect how many videos are assigned to a wrong node ID. Mathematically, the Hamming distance between a prediction and its corresponding ground truth is $\sum_{i=1}^{N}[a_i \neq \hat{a}_i]$. The average Hamming distance for our example is $\frac{3}{10}$, since there are three videos, *i.e.*, $v_5$, $v_6$, and $v_9$, are assigned to the incorrect node IDs.

**Metric 3: Video-level Euclidean distance.** Although the above video-level Hamming Distance can show how many videos are wrongly assigned, it is still unclear how much the wrongly predicted node IDs deviate from the ground truth node IDs. Thus, we propose video-level Euclidean distance, which considers the absolute distance that a wrongly predicted video is shifted from its ground truth node IDs. It can capture the magnitude of errors in the video order. Formally, the Euclidean distance between a prediction and its corresponding ground truth is calculated as $\sum_{i=1}^{N}|a_i - \hat{a}_i|$. Obviously, the above video-level Hamming distance is a lower bound of this video-level Euclidean distance. In the given timeline example, the average Euclidean distance is $\frac{3}{5}$, because the absolute error distance of $v_5$, $v_6$, and $v_9$ are all $2$.

**Metric 4: Video pairwise agreement accuracy.** To provide a more comprehensive evaluation of the video timeline modeling, we further include a video pairwise agreement accuracy metric. This metric is a straightforward extension of Kendall tau ranking distance [Kendall, 1938] by further considering the fact that multiple videos could have the same ranking. Specifically, we compute the number of pairwise video orders that are correctly predicted out of all possible pairwise orders. Note that for any pair of videos $(v_i, v_j)$, there are three possible order relationships, including $a_i < a_j$, $a_i = a_j$, and $a_i > a_j$. This metric complements the other metrics by focusing on the pairwise relationships between videos. In our example with $10$ videos, there are $45$ possible pairwise orders, and $32$ of them are correctly predicted, so the pairwise agreement accuracy for this timeline is $\frac{32}{45}$.

For all the above four metrics, we can compute both macro and micro averages over the dataset. Macro averaging gives equal weight to each timeline, while micro averaging gives equal weight to each node (for the node-level precision and recall metric), each video (for video-level Hamming distance and Euclidean distance metrics), or each video pair (for video pairwise agreement accuracy).

# 4 Approaches

The benchmark proposed in Section 3 provides a standardized testbed for future research in this new direction. Here, we propose the following three exploratory approaches to tackle the defined video timeline modeling problem. We anticipate these models can serve as baselines for future research.

Given that each video in a timeline needs to be assigned to a specific node ID and the maximum number of nodes on the timelines in our YouTube-News-Timeline dataset is $24$, in our following approaches, we formulate the model to perform a 24-class classification task to predict the node IDs. More specifically, the $k$-th class, for $k = 1, 2, \ldots, 24$, corresponds to the $k$-th node ID. Since learning from such a numerous number of raw videos is computationally expensive and raw video data often contains a large amount of noise and redundancy, we use an in-house feature extractor to obtain video embeddings for the following models. We anticipate that using other publicly available

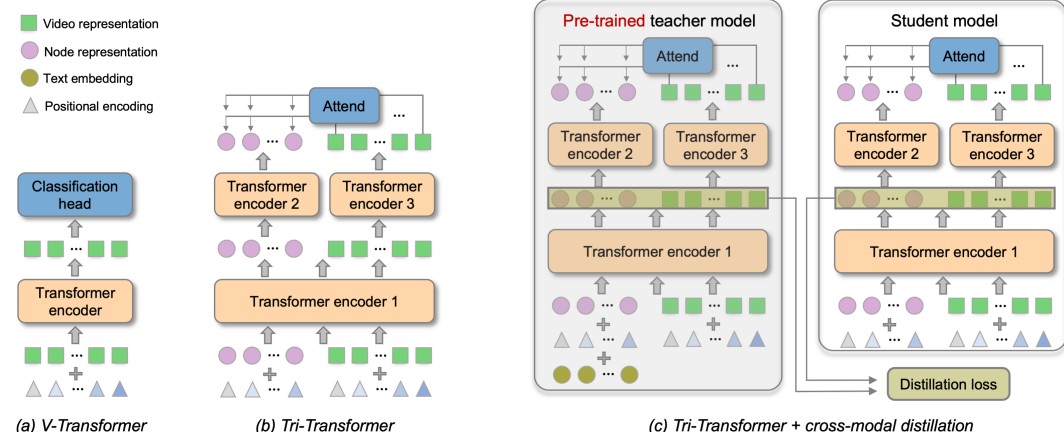

*(a) V-Transformer*  *(b) Tri-Transformer*  *(c) Tri-Transformer + cross-modal distillation*

Figure 6: Illustration of our proposed approaches, including (a) V-Transformer, (b) Tri-Transformer, and (c) Tri-Transformer + cross-modal distillation.

video feature extractors, such as MediaPipe YouTube-8M feature extractor[1], can achieve similar performance and does not affect the main trend of our experimental results. We encourage to use the same featurization strategy for model comparison. It is worth noting that the input videos for a news topic can be considered as a set without any specific order. However, in our proposed models, we consider ordering the videos by their release time, as we have found through our ablation study that this prior knowledge is beneficial for improving performance significantly.

**V-Transformer.** Our first model uses Transformer to encode input videos. Specifically, the embeddings of input videos are ordered by the release time of the videos and such order information is incorporated into the model via positional encoding [Vaswani et al., 2017]. Such a Transformer encoder aims to learn the evolution dependencies among videos. The resulting representations are then fed into a fully connected layer to perform 24-class classification, where each class corresponds to a specific node ID on the timeline. The whole model is trained by the cross entropy loss. An illustration of this model is shown in Figure 6(a).

**Tri-Transformer.** The main limitation of V-Transformer is that the 24 classes are treated independently. However, the nodes on a timeline are ordered and their underlying dependencies should be explicitly captured. Motivated by this, we further propose to model nodes as learnable embeddings, and they are ordered according to the node order. As shown in Figure 6(b), the randomly initialized and learnable node embeddings and video embeddings are concatenated into a single sequence, which is the input for the first Transformer encoder. Note that they use two separate sets of positional encodings to indicate the node order and video release time order, respectively. Intuitively, the first Transformer encoder aims at capturing the interactions among all tokens, including node tokens and video tokens. With this module, the obtained node representations are dependent on the input video embeddings. In other words, the obtained node representations are customized for each input video set, which is intuitively more effective than video-set-agnostic node embeddings for the subsequent node assignment process. We further use two additional Transformer encoders to model the node-node interactions and video-video interactions, respectively. Next, we attend each video representation to all node representations. The attention score indicates the probability that the video belongs to the corresponding node. Inspired by pointer networks [Vinyals et al., 2015], such scores are then used by the cross entropy loss to train the whole model. Our model is named as Tri-Transformer since it consists of three Transformer encoders.

**Tri-Transformer + cross-modal distillation.** In our labeled video-associated timelines, as shown in Figure 3, we also have the text description for the nodes in addition to the assignments from videos to nodes. Such texts contain rich semantics of the nodes and are supposed to be informative for assigning videos to nodes. However, these privileged text descriptions are not available during inference but training only. These considerations raise a natural question. *Is it possible to use the rich text information during training to improve inference performance?*

---

[1] https://research.google.com/youtube8m

Table 2: Performance of the proposed methods on our benchmark in terms of node-level precision and recall, video-level Hamming distance, video-level Euclidean distance, and video pairwise agreement accuracy. ↑ (↓) denotes that higher (lower) value indicates better performance. The top two results in terms of each metric are highlighted as **1st** and 2nd.

| Method | Precision/Recall↑ | | Hamming distance↓ | | Euclidean distance↓ | | Agreement accuracy↑ | |
|---|---|---|---|---|---|---|---|---|
| | *Macro* | *Micro* | *Macro* | *Micro* | *Macro* | *Micro* | *Macro* | *Micro* |
| V-Transformer | 68.8%/68.9% | 64.4%/63.0% | 0.477 | 0.519 | 0.765 | 0.916 | 84.2% | 93.8% |
| Tri-Transformer | 71.4%/**71.2%** | 66.6%/65.6% | 0.453 | 0.490 | 0.716 | 0.837 | **85.4%** | 94.3% |
| Tri-Transformer + cross-modal distillation | **72.5%**/71.1% | **68.4%**/**65.9%** | **0.436** | **0.470** | **0.684** | **0.804** | 85.2% | **94.5%** |
| A reference optimum | 79.8%/78.3% | 75.0%/72.9% | 0.308 | 0.362 | 0.380 | 0.473 | 88.8% | 95.5% |

To answer this question, we propose to use knowledge distillation techniques [Hinton et al., 2015, Gou et al., 2021]. Specifically, as illustrated in Figure 6(c), we first train a teacher model, *i.e.*, a Tri-Transformer model, that includes text embeddings as input to capture semantic knowledge contained in the node text descriptions. The text embeddings are still given by the pre-trained NewsEmbed model [Liu et al., 2021], and are added to the corresponding learnable node embeddings as input to the Tri-Transformer. The parameters of our teacher model are fixed after the training. Our student model is also a Tri-Transformer model, without text embeddings as input. To train the student model, in addition to the regular cross entropy loss, we have another knowledge distillation loss defined as the $\mathcal{L}_2$ distance of the node and video representations between the student model and the pre-trained teacher model. We specifically focus on the node and video representations generated by the first Transformer encoder, as those in the teacher model contain the most essential information learned from the text embeddings and their interactions with videos. The final loss function is a weighted summation of the cross entropy loss and the distillation loss. After the training, only the student model is used for inference. The above knowledge distillation strategy transfers the learned text semantics knowledge from the teacher model to the student model, which does not receive text embeddings as input. Such distillation is cross-modal [Gupta et al., 2016] since the text modality is not available during inference.

Since we convert the timeline modeling problem to a classification problem in our approaches, there might end up with empty classes in our prediction results, thus leading to intermediate empty nodes on the timeline. To address this issue, we apply a post-processing step to skip those intermediate empty nodes. For instance, if the predicted node IDs for a 4-video input are $\{1, 1, 4, 2\}$, we refine it to $\{1, 1, 3, 2\}$ by skipping the original third node, thereby forming a consecutive timeline. We leave the study of a more integrated approach for future work, as discussed in Section 6.

## 5 Experiments

**Setup.** We use 2 Transformer layers for all Transformer encoders in our models. In our third method, we set the weight for the cross entropy loss and the distillation loss to be 1 and 0.1, respectively. We train all models for 100 epochs with the Adam optimizer [Kingma and Ba, 2015] and a batch size of 32. We did not tune the training hyperparameters extensively. The key hyperparameters learning rate and dropout rate are respectively selected from $\{0.01, 0.001, 0.0005\}$ and $\{0, 0.1, 0.25, 0.5\}$ based on the validation performance.

**Overall results.** The test performance of our proposed three approaches on YouTube-News-Timeline is present in Table 2. Although these three models are exploratory attempts at solving the video timeline modeling problem, they perform fairly well in terms of the correct identification of events and the correct evolution ordering of videos. For example, based on the micro average results, the models were able to recall over 60% of event nodes and correctly infer over 90% of video

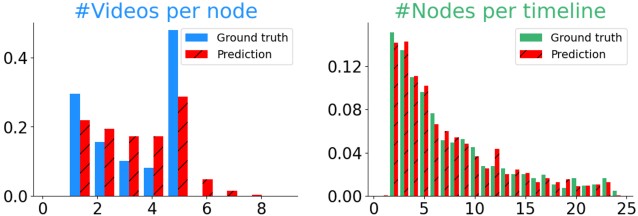

Figure 7: Comparison of distributions of the number of videos per node and the number of nodes per timeline between ground truth and predictions given by Tri-Transformer over the testing subsets.

pairwise orders. In addition, the micro-average results of video-level Euclidean distance demonstrate

Table 3: Results of ablation studies. The table schema follows Table 2.

| Method | Precision/Recall↑ | | Hamming distance↓ | | Euclidean distance↓ | | Agreement accuracy↑ | |
|---|---|---|---|---|---|---|---|---|
| | *Macro* | *Micro* | *Macro* | *Micro* | *Macro* | *Micro* | *Macro* | *Micro* |
| Tri-Transformer | 71.4%/71.2% | 66.6%/65.6% | 0.453 | 0.490 | 0.716 | 0.837 | 85.4% | 94.3% |
| w/o video PE | 50.1%/47.7% | 43.4%/40.4% | 0.621 | 0.647 | 1.523 | 2.105 | 58.3% | 72.1% |
| w/o video PE and encoder 2&3 | 47.5%/43.3% | 39.9%/37.7% | 0.638 | 0.656 | 1.576 | 2.165 | 54.8% | 72.0% |

that, on average, the predicted node IDs deviate less than one absolute position from the ground truth node IDs. A comparison of the performance between V-Transformer and Tri-Transformer shows that explicitly modeling nodes and their underlying dependencies and interactions with videos can consistently achieve more effective performance. Furthermore, using knowledge distillation via Tri-Transformer + cross-model distillation can further boost performance by transferring the learned textual semantics from the teacher model to the student model.

Aside from quantitative results, we examine the distributions of the number of videos per node and the number of nodes per timeline between ground truth and predictions given by Tri-Transformer. As shown in Figure 7, such distributions of predicted results closely match those of the ground truth. It is important to note that the dataset sets a limit of five videos per node, but we do not enforce this constraint during prediction, and we rely solely on the model's raw predictions. This explains why our model sometimes assigns more than five videos to a single node. The visualizations of several predicted timelines are included in the appendix.

**An optimum bound study.** Our cross-modal distillation strategy involves using node texts as input in the teacher model and transferring the learned knowledge to the student model that does not have such input. This is because we assume that such textual node information is not available during inference. Given that we do have such textual node information in our test set of YouTube-News-Timeline dataset, we can evaluate the test performance of the teacher model that includes text information as input and treat such performance as a reference optimum of our Tri-Transformer model without using textual information. The results included in Table 2 reveal that, while our cross-modal distillation approach does improve the performance over Tri-Transformer, there is still an obvious performance gap compared to the reference optimum. Therefore, developing more effective ways to incorporate textual information during training is a potential future direction, as discussed in Section 6.

**Ablation studies.** We further perform ablation study experiments based on Tri-Transformer to evaluate two key designs. First, we analyze the effect of incorporating the prior knowledge of video release time. As described in Section 4, we use positional encodings (PE) to indicate the order of input videos according to their release time. In our ablation model, we remove such positional encodings and treat the input videos as a set instead. The results shown in Table 3 clearly demonstrate that this order information is a strong prior knowledge and significantly improves the performance. Additionally, we further remove Transformer encoders 2&3 and observe a decrease in performance, highlighting the importance of modeling the node-node interactions and video-video interactions.

## 6   Conclusion and Outlook

In this work, we introduce a novel and important video timeline modeling task, which constructs a video-associated timeline from a set of videos, to help understand unstructured news videos. We provide a standardized testbed for this new task with a realistic dataset and a comprehensive evaluation protocol. We further develop and evaluate several intuitive approaches as simple baselines for future studies. The proposed benchmark dataset and methods will open the door for future research in video timeline modeling.

Here, we highlight several promising future directions in this field, including more principle exploration for problems and methodologies. On one hand, more effective and principle methodologies are highly desired. First, as demonstrated in the experiments, we can develop more effective strategies to leverage textual information during training. Second, while our current method, as an exploratory work, treats the problem as a multi-class classification problem, it would be more principle to consider the problem as a ranking problem and use differentiable ranking models. Third, our algorithms assign a set of videos simultaneously, which may result in different assignments for the same video if we add or delete a video in the set. It remains challenging to model video interactions while keeping the prediction of videos less dependent. This aspect presents a potential avenue for enhancement.

In addition, future research can also be made from a problem-oriented perspective. Given that our defined problem and evaluation standard concentrate solely on assigning input video sets to ordered nodes, one possible extension is to integrate the event summarization step with our defined problem to construct a timeline with both associated videos and event information. The event information can be in the form of text, key frames, *etc*. Although our YouTube-News-Timeline dataset can be used to evaluate the generated event text descriptions, the evaluation protocol and algorithms need to be dedicatedly redesigned. In addition, extending the single linear timeline to more complex relationship modeling, such as multiple timelines [Yu et al., 2021] and graphs, can be a promising future direction to enhance the understanding of news stories

## Disclaimer

The reference timelines used to construct the dataset are crawled from the web, and the videos are sourced from YouTube. The opinions expressed in the timelines and videos do not necessarily reflect our own, and we do not endorse or promote any specific viewpoint. The dataset is intended for research and educational purposes only, and users should exercise their own judgment when interpreting and using it.

## Acknowledgments

We would like to express our gratitude to Anish Nair from Google who initiated and sponsored this project and as well as consistently provided constructive discussion throughout the delivery of the paper. We also thank the reviewers for their constructive feedback, which improved the quality of this work.

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

# A  License

The YouTube-News-Timeline dataset is under the CC BY 4.0 International license. Please refer to `https://creativecommons.org/licenses/by/4.0/legalcode` for license details.

# B  Data Format Example

A data format example is shown below.

Listing 1: A data example

```
1  {"https://apnews.com/article/japan-accidents-tsunamis-earthquakes-42
       ↪d4947609becd7f141e9524a8c98937":  # The URL link of the webpage
       ↪where we crawl the timeline.
2      [
3        [
4          "OhEbGK4PnZg",
5          "cl19tfn33hI",
6          "ROl6z0HaUAM",
7          "5QhCsR-t-qM",
8          "ev3FBIoHMX8"
9        ],
10        [
11          "psAuFr8Xeqs",
12          "BsRd7WQuBHc",
13          "Dp_8rLL1Y18",
14          "h1m7GFPAq3o"
15        ],
16        [
17          "f4TaKPKe1gg",
18          "DLlsKd-QC2o"
19        ],
20        [
21          "ocluW1Vhvcg",
22          "vusthiUFx_0",
23          "vGHzuZQLYtg",
24          "7XpLbhQxpLw",
25          "UsPFUzXisq4"
26        ],
27        [
28          "hA3fNK0rxcs"
29        ]
30      ] # The URL links of the retrieved YouTube news videos. Each list
            ↪in the nested list corresponds to one node on the timeline.
            ↪These nodes are ordered in the nested list.
31  }
```

# C  More Dataset Characteristics

We include more details on the dataset's characteristics here. As demonstrated in Figure 8, the reference timelines are curated from a diverse set of more than $1,000$ publishers. The primary topics, each recurring more than 30 times, are depicted in In Figure 9. The topic annotations are given by an in-house proprietary entity linking algorithm. Moreover, the distribution of the event date, corresponding to each node on timelines, is shown in Figure 10.

# D  More Experimental Studies

To assess the impact of video duration on our model's performance, in Figure 11, we present a correlation between the video length and the corresponding video-level Euclidean distance, based on our Tri-Transformer model over the test set. The video-level Euclidean distance is not sensitive

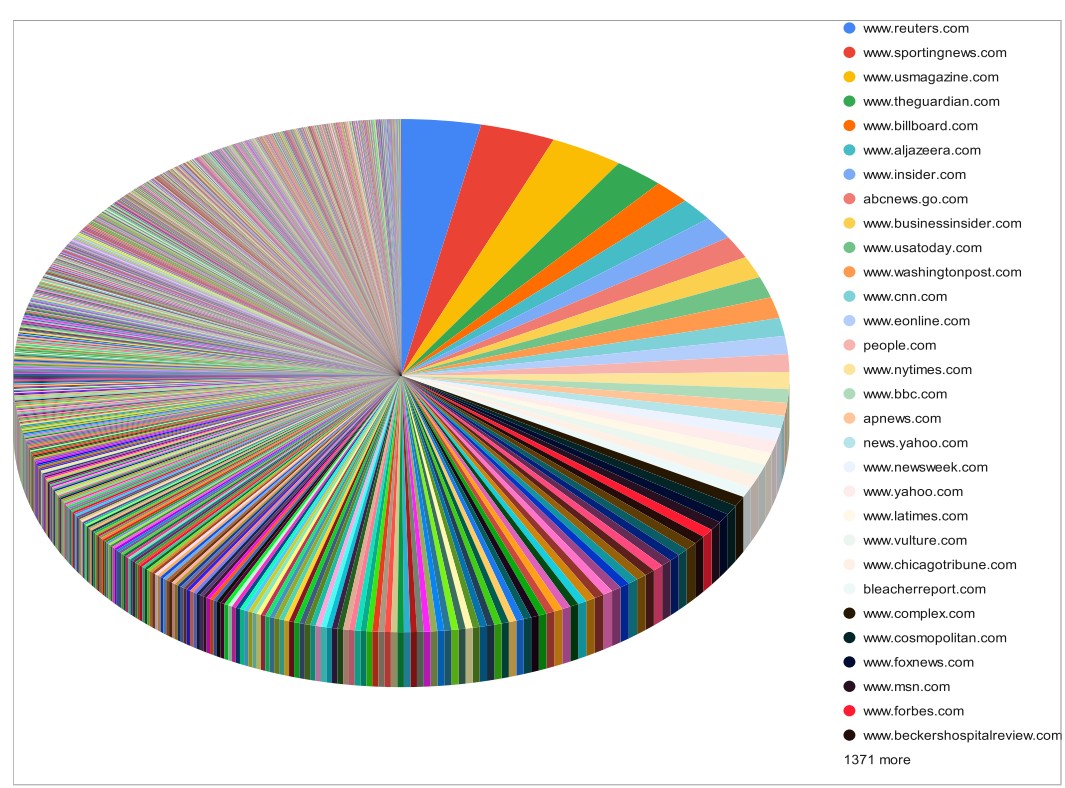

Figure 8: Pie chart of the news publishers.

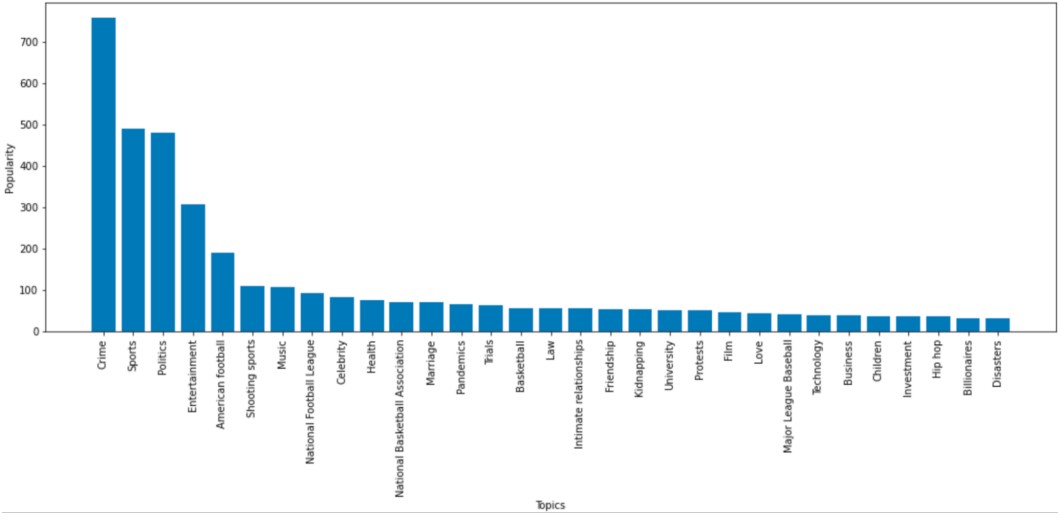

Figure 9: Distribution of the covered main topics.

to video length. It can be observed that the variance becomes large for longer videos as the longer videos are less in both the training and test sets.

## E   Examples of Predicted Timelines

We present two examples of timelines predicted by our Tri-Transformer model in Table 4 and 5. The videos that have been assigned to incorrect nodes are highlighted for clarity.

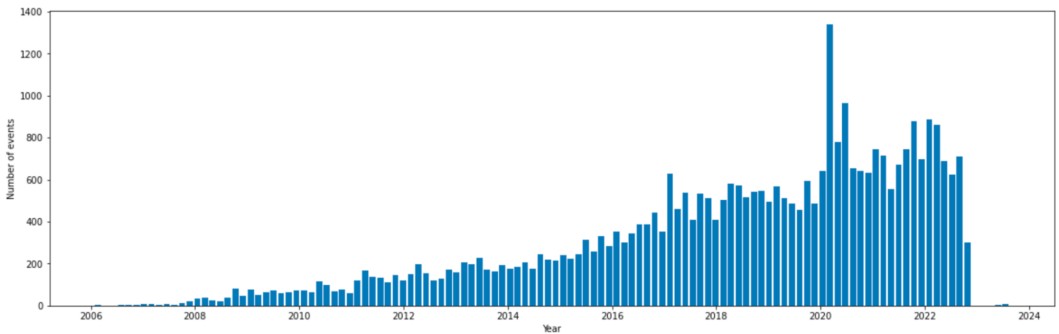

Figure 10: Distribution of the event date.

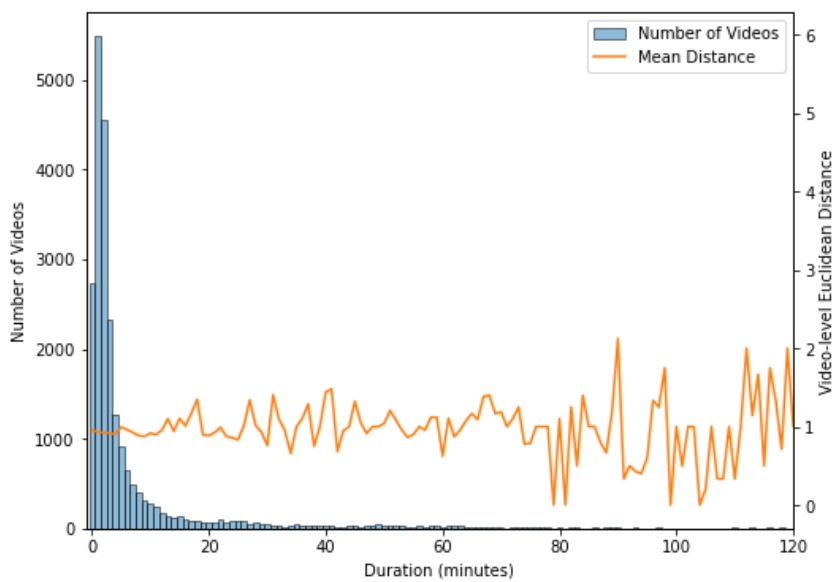

Figure 11: Number of videos and average video-level Euclidean distances in relation to video duration.

| | | |
|---|---|---|
| #ground truth nodes = 8, #predicted nodes = 8, #videos = 29 | | |
| node-level precision/recall = 100.0%/100.0%, video-level Hamming distance = 0.103, video-level Euclidean distance = 0.103, video pairwise agreement accuracy = 96.6% | | |
| **Node ID** | **Ground truth videos (titles)** | **Predicted videos (titles)** |
| 1 | tT_AWx7dgjg ("China Rocket Launches, Spacewalk Next")
YJrnfvQZNYo ("China Launches Its Third Manned Space Mission") | tT_AWx7dgjg ("China Rocket Launches, Spacewalk Next")
YJrnfvQZNYo ("China Launches Its Third Manned Space Mission") |
| 2 | UlcL9aXpOYE ("LIFT OFF: China launches space module rocket Tiangong 1")
8zK8jHpLV9c ("China launches first module for space station")
2CU0RKMW2n8 ("China launches first stage of space station plan") | UlcL9aXpOYE ("LIFT OFF: China launches space module rocket Tiangong 1")
8zK8jHpLV9c ("China launches first module for space station") |
| 3 | MAx8UrCm5Cg ("SPACE MISSION: China's Shenzhou-8 carries out first docking")
Tj1zGxShID4 ("Success as China passes its first space docking test") | 2CU0RKMW2n8 ("China launches first stage of space station plan")
MAx8UrCm5Cg ("SPACE MISSION: China's Shenzhou-8 carries out first docking")
Tj1zGxShID4 ("Success as China passes its first space docking test") |
| 4 | 3A6pvG676aM ("China Lands Spacecraft on Moon")
l-XYRQWGEZM ("China's Chang'e3 unmanned spacecraft lands on the moon")
BxdBFCwqlGw ("State TV in China broadcasts Moon landing")
oEHniKlucL0 ("China lands rover on the moon")
PGrf1_lpjH0 ("Chinese lunar rover lands on moon") | 3A6pvG676aM ("China Lands Spacecraft on Moon")
l-XYRQWGEZM ("China's Chang'e3 unmanned spacecraft lands on the moon")
BxdBFCwqlGw ("State TV in China broadcasts Moon landing")
oEHniKlucL0 ("China lands rover on the moon")
PGrf1_lpjH0 ("Chinese lunar rover lands on moon") |
| 5 | RodztOlgZW0 ("China launches Tiangong-2 space lab")
euBlfv9S-ZE ("China successfully launches 2nd space laboratory Tiangong-2")
goRzJM4PlEI ("China successfully launches Tiangong-2 space lab")
Dw9RHNVmswE ("The launch of Tiangong-2 space lab")
1stYgsnT9Zs ("Cina, lanciato in orbita il laboratorio spaziale Tiangong-2") | RodztOlgZW0 ("China launches Tiangong-2 space lab")
euBlfv9S-ZE ("China successfully launches 2nd space laboratory Tiangong-2")
goRzJM4PlEI ("China successfully launches Tiangong-2 space lab")
Dw9RHNVmswE ("The launch of Tiangong-2 space lab")
1stYgsnT9Zs ("Cina, lanciato in orbita il laboratorio spaziale Tiangong-2") |
| 6 | kQrLjgvIXEc ("Chinese probe Chang'e 4 lands on far side of moon")
A2AXeRN-7Uk ("China's Chang'e-4 lands on moon's far side")
WpzJblVRdO4 ("China lands Chang'e-4 space probe on the far side of the moon")
apVqFquXYIg ("Chang'e-4 Probe Takes Panoramic Photos on Moon's Far Side")
iEThGnpwAxU ("Chang'e-4 Probe Takes First Photo of Moon's Far Side") | kQrLjgvIXEc ("Chinese probe Chang'e 4 lands on far side of moon")
A2AXeRN-7Uk ("China's Chang'e-4 lands on moon's far side")
WpzJblVRdO4 ("China lands Chang'e-4 space probe on the far side of the moon")
apVqFquXYIg ("Chang'e-4 Probe Takes Panoramic Photos on Moon's Far Side")
iEThGnpwAxU ("Chang'e-4 Probe Takes First Photo of Moon's Far Side") |
| 7 | J7M-nEJnr4k ("China puts final satellite for Beidou into orbit")
JCa9UPHMaWE ("China launches 100% domestically produced final satellite to complete Beidou system") | J7M-nEJnr4k ("China puts final satellite for Beidou into orbit")
JCa9UPHMaWE ("China launches 100% domestically produced final satellite to complete Beidou system")
WSkCBaN_CLM ("China launches unmanned probe in first independent Mars mission")
fwuWTd0NScc ("China launches its first unmanned mission to Mars") |
| 8 | WSkCBaN_CLM ("China launches unmanned probe in first independent Mars mission")
fwuWTd0NScc ("China launches its first unmanned mission to Mars")
sCiwZTnxBvo ("Tianwen 1: China launches first independent mission to Mars")
UqzBbubHtvc ("Beijing launches unmanned interplanetary mission to Mars")
0OCPNfGrrcQ ("China launches unnamed Mars probe") | sCiwZTnxBvo ("Tianwen 1: China launches first independent mission to Mars")
UqzBbubHtvc ("Beijing launches unmanned interplanetary mission to Mars")
0OCPNfGrrcQ ("China launches unnamed Mars probe") |

Table 4: A timeline of "Major milestones in Chinese space exploration".

| #ground truth nodes = 10, #predicted nodes = 12, #videos = 40 | |
| --- | --- |
| node-level precision/recall = 58.3%/70.0%, video-level Hamming distance = 0.525, video-level Euclidean distance = 0.575, video pairwise agreement accuracy = 91.2% | |

| Node ID | Ground truth videos (titles) | Predicted videos (titles) |
| --- | --- | --- |
| 1 | aUs-BkqMLAM ("Schools Respond To U.S. Department of Ed Letter on Transgender Students")
oZpYA8v5PRI ("Local reaction to school transgender guidelines")
w7jeruoR0PM ("Local school divisions, and the public, react to transgender announcement") | aUs-BkqMLAM ("Schools Respond To U.S. Department of Ed Letter on Transgender Students")
oZpYA8v5PRI ("Local reaction to school transgender guidelines") |
| 2 | rghWbw4DKwg ("Protections pulled from transgender school restrooms")
TNmDduNrsZY ("How scrapping transgender bathroom guidelines impacts schools")
kPFx_FkEx2s ("Rights and Recourse, 26 February 2017")
za2hWMaef6U ("CI Bitesize: Trans bathroom guidance revoked") | w7jeruoR0PM ("Local school divisions, and the public, react to transgender announcement")
rghWbw4DKwg ("Protections pulled from transgender school restrooms")
TNmDduNrsZY ("How scrapping transgender bathroom guidelines impacts schools")
kPFx_FkEx2s ("Rights and Recourse, 26 February 2017") |
| 3 | rX9qHJngRPk ("Vanderbilt kicker Sarah Fuller becomes first woman to score in a Power 5 game")
-8MfQM9PAxk ("Sarah Fuller Becomes First Female Player To Kick Extra Point")
IfgwDHOHbCI ("Tennessee vs. Vanderbilt: Sarah Fuller makes history")
9CaIF7nY6F8 ("Women's Football Highlight Show") | za2hWMaef6U ("CI Bitesize: Trans bathroom guidance revoked")
rX9qHJngRPk ("Vanderbilt kicker Sarah Fuller becomes first woman to score in a Power 5 game") |
| 4 | j8kudE06p08 ("President Biden Delivers Remarks and Signs Executive Orders")
5xlSQ34b5bA ("President Joe Biden signs executive orders as 1st official act")
IX9nMUn2lIY ("Biden delivers remarks, signs exec order on racial equity agenda")
NTus5FevWhM ("President Joe Biden signs his first executive orders")
Es_SMG8b3sI ("Unpacking Biden's first Executive Orders") | -8MfQM9PAxk ("Sarah Fuller Becomes First Female Player To Kick Extra Point")
IfgwDHOHbCI ("Tennessee vs. Vanderbilt: Sarah Fuller makes history")
9CaIF7nY6F8 ("Women's Football Highlight Show")
j8kudE06p08 ("President Biden Delivers Remarks and Signs Executive Orders") |
| 5 | IDkDrTvxsLY ("Presidential address: Watch Biden's full speech from March 11, 2021")
tjVaIdsYekk ("Education announcement - March 12, 2021") | 5xlSQ34b5bA ("President Joe Biden signs executive orders as 1st official act")
IX9nMUn2lIY ("Biden delivers remarks, signs exec order on racial equity agenda")
NTus5FevWhM ("President Joe Biden signs his first executive orders")
Es_SMG8b3sI ("Unpacking Biden's first Executive Orders")
IDkDrTvxsLY ("Presidential address: Watch Biden's full speech from March 11, 2021") |
| 6 | 3s9He2qdb8w ("Supreme Court rules in favor of transgender student Gavin Grimm")
ZvYRB7kAfpE ("Supreme Court refuses case for transgender bathroom rule") | tjVaIdsYekk ("Education announcement - March 12, 2021") |
| 7 | 8GwdG2qunIY ("U.S. women soccer players reach $24 million settlement in fight for equal pay")
HCSkjUT6tH0 ("U.S. Soccer and women's national team players settle equal pay lawsuit for $24 million")
zMxP6HxfKqU ("US women soccer players settle equal pay suit for $24M")
EAZB29f8w5k ("US Women's Soccer players reach landmark $24 million settlement with Soccer Federation")
3--vxNT1jrU ("'A win for everyone:' U.S. women's soccer settles equal pay dispute") | 3s9He2qdb8w ("Supreme Court rules in favor of transgender student Gavin Grimm")
ZvYRB7kAfpE ("Supreme Court refuses case for transgender bathroom rule")
8GwdG2qunIY ("U.S. women soccer players reach $24 million settlement in fight for equal pay")
HCSkjUT6tH0 ("U.S. Soccer and women's national team players settle equal pay lawsuit for $24 million")
zMxP6HxfKqU ("US women soccer players settle equal pay suit for $24M")
3--vxNT1jrU ("'A win for everyone:' U.S. women's soccer settles equal pay dispute") |
| 8 | kAnbWxtiddA ("Lia Thomas Becomes 1st Transgender Athlete To Win NCAA championship")
jHXgde2IZJg ("Reaction to trans swimmer Lia Thomas' NCAA Championship")
HLCNg4ISygU ("Making Waves: Transgender Swimmer Lia Thomas' NCAA Win Sparks Public Protest")
i5pvCVL1DNk ("Lia Thomas competes in Division")
rZj76yicEaE ("WATCH: Lia Thomas Swims 4:33.82 in 500 Free Prelims at NCAA Championships") | EAZB29f8w5k ("US Women's Soccer players reach landmark $24 million settlement with Soccer Federation")
kAnbWxtiddA ("Lia Thomas Becomes 1st Transgender Athlete To Win NCAA championship")
jHXgde2IZJg ("Reaction to trans swimmer Lia Thomas' NCAA Championship") |
| 9 | 2IS2WYD9fk8 ("2021-22 Women's Basketball Season Recap")
a8rsZikLagE ("Locked on Women's Basketball, March 24, 2022: 68 to the Sweet 16!")
aV0yBBopnnI ("NCAA Women's Basketball Tournament comes to Spokane and other top stories at 8 a.m.")
AzqRk3HOeK8 ("Women's Basketball Weekly Update (03.29.2022)")
K6Z7UsHUe_k ("NAU Women's Basketball Championship Update") | HLCNg4ISygU ("Making Waves: Transgender Swimmer Lia Thomas' NCAA Win Sparks Public Protest")
i5pvCVL1DNk ("Lia Thomas competes in Division")
rZj76yicEaE ("WATCH: Lia Thomas Swims 4:33.82 in 500 Free Prelims at NCAA Championships")
2IS2WYD9fk8 ("2021-22 Women's Basketball Season Recap")
a8rsZikLagE ("Locked on Women's Basketball, March 24, 2022: 68 to the Sweet 16!")
aV0yBBopnnI ("NCAA Women's Basketball Tournament comes to Spokane and other top stories at 8 a.m.")
K6Z7UsHUe_k ("NAU Women's Basketball Championship Update") |
| 10 | XHEAZkiObYw ("UConn vs. South Carolina \| Full Game Highlights")
tuDTa9On1IA ("South Carolina vs. UConn - Women's NCAA tournament championship highlights")
hb7e6vh71p4 ("South Carolina vs. UConn: 2022 NCAA women's national championship")
zoaApnfmIqc ("Press Conference: South Carolina vs. UConn Postgame - 2022 NCAA Tournament")
JXpzJ20jJZY ("South Carolina women's basketball national championship celebration") | AzqRk3HOeK8 ("Women's Basketball Weekly Update (03.29.2022)")
XHEAZkiObYw ("UConn vs. South Carolina \| Full Game Highlights") |
| 11 | | tuDTa9On1IA ("South Carolina vs. UConn - Women's NCAA tournament championship highlights")
hb7e6vh71p4 ("South Carolina vs. UConn: 2022 NCAA women's national championship") |
| 12 | | zoaApnfmIqc ("Press Conference: South Carolina vs. UConn Postgame - 2022 NCAA Tournament")
JXpzJ20jJZY ("South Carolina women's basketball national championship celebration") |

Table 5: A timeline of "50 Years of Title IX: The Defining Moments of Women's Sports".

# F Societal Impacts

We expect that the proposed benchmark dataset and methods will facilitate future advancements in video timeline modeling. On the positive side, it offers a helpful tool for understanding and navigating large volumes of news video data, enabling more efficient news consumption and ensuring a more comprehensive understanding of events.

One the negative side, the crawled timelines might not always reflect absolute precision. They might be mistakenly used as evidence in some situations. This highlights the importance of using and interpreting these timelines carefully. In addition, while we have taken steps to diversify our data sources, news content can inherently carry biases based on various factors. We acknowledge this challenge and emphasize the future need for more diverse data sourcing to capture a broad spectrum of perspectives and reduce inherent biases. Also, there could be potential misuse if the technology were applied unethically. Specifically, the construction of timelines could be manipulated to present events in a way that supports a particular opinion, thereby distorting the truth. To alleviate this potential concern, regular validation and fact-checking mechanisms can help ensure the constructed timelines align with factual occurrences.

