# OpenReview forum: "Video Timeline Modeling For News Story Understanding"
_NeurIPS.cc/2023/Track/Datasets_and_Benchmarks — NeurIPS 2023 Datasets and Benchmarks Spotlight_

### Official Review · Reviewer_xNvC · 2023-07-22
**Good paper, more clarification about the dataset is required**

**Rating:** 6
**Confidence:** 3
**Correctness:** I provided my comments in the opportu…

**Strengths:**

I read the paper with great interest and find it well-written and describes the objectives and methods clearly. It describes a new application/task on news video and also present the dataset that can be used for this purpose.

**Additional Feedback:**

I provided my comments/suggestions in the previous sections.

**Clarity:**

Yes, the paper is well written, but needs more clarification about the data curation process (Please see my comments in  the opportunities for improvement section.

**Documentation:**

I did not find this documentation for this paper.

**Ethics:**

As these are news data, I don't think there is privacy concern, but there might be concern about the licensing. The authors should explain that in the paper.

**Limitations:**

I did not see a discussion on the societal impact in the paper, other than one part that the authors made disclaimer about the model's timeline alignment and this might not be the actual timeline. The author present a section in the supplementary file about this. The authors could expand on this, maybe describing that because the groundtruth timelines were also obtained by web crawling, they may not be precise enough, and a potential wrong prediction of news alignment may indeed have negative societal impact. For example, it might be used as evidence, while it is not indeed a confident reference.

**Opportunities For Improvement:**

While the authors present the video timeline modeling as a new benchmark task which is interesting and may have applications in news industry, it seems highly related to video recommendation tasks. It is basically comes to recommending videos for each node in the timeline.

For the curated dataset, it is unclear how the news topic have been selected? how the videos related to each topic is selected? what time interval was chosen for the each news topic? What topics over what period of time were included in the dataset? What are the statistical characteristics of the dataset? What information is included to create the timeline (other than the time stamp)? Was location of the event a factor? So generally, the authors should provide more clear information about how the data was curated, what was the rationale of choices being made, how the data was cleaned, and what was the statistical characteristics of the final dataset?



**Relation To Prior Work:**

More clarification on the difference of their proposed task with recommendation systems is required.

**Summary And Contributions:**

This paper introduces Video Timeline Modeling, a task to align news related videos to an inferred or pre-defined timeline of events. The authors also curate a benchmark dataset which can be used for the defined task. Finally, they present a set of metrics and frameworks that can be used for this task.

---

> ### Author Response · Authors · 2023-08-23
> **Clarification with relate work, and more dataset information**
>
> Thank you for your constructive comments.
>
> > `While the authors present the video timeline modeling as a new benchmark task which is interesting and may have applications in news industry, it seems highly related to video recommendation tasks. It is basically comes to recommending videos for each node in the timeline.`
>
> As described at the end of Section 3.1, in our problem setting, the textual timeline is not given as input. This setup aligns with the realistic scenario where a well-organized and timely timeline is typically unavailable for news, particularly for breaking news, but we are able to accurately retrieve the videos related to a news topic based on keyword filters or embedding guided retrieval. ​​ We acknowledge that another setting, as mentioned by the reviewer, where videos are recommended or assigned for each node based on a predefined timeline could have its merits. Both setups are valid, each stemming from distinct problem assumptions and targeting different real-world situations. We further clarified this in our revision.
>
>
> > ​​`For the curated dataset, it is unclear how the news topic have been selected? how the videos related to each topic is selected? what time interval was chosen for the each news topic? What topics over what period of time were included in the dataset? What are the statistical characteristics of the dataset? What information is included to create the timeline (other than the time stamp)? Was location of the event a factor? So generally, the authors should provide more clear information about how the data was curated, what was the rationale of choices being made, how the data was cleaned, and what was the statistical characteristics of the final dataset?`
>
>
> Thank you for your inquiries regarding the dataset details. We value transparency in research and will provide as much detail as possible.
>
> (1) We crawled news timelines from a diverse set of news sources. As shown in the added [Figure 8](https://ibb.co/sKKF26x) in the supplementary material, our reference timelines are sourced from various publishers, ensuring diversity and minimizing inherent biases towards any particular source. (2) As described in the data collection pipeline (Figure 2), the videos for each node are obtained with a high precision by an embedding-based retrieval technique. (3) The time interval was not specifically chosen but determined by the news itself. We added [Figure 10](https://ibb.co/s3rNwmz) in the supplementary material to show the distribution of event dates on the timelines in our dataset. The timelines cover an expansive temporal range from 2006 to 2023. (4) We added [Figure 9](https://ibb.co/VVqrLqs) in the supplementary material to show the covered primary topics, which are quite diverse. (5) Due to potential intellectual property concerns, we provide the URLs corresponding to the textual timelines in our dataset. Users can crawl the time stamps, textual descriptions, and other relevant information (such as locations if applicable) if needed. For our formulated video timeline modeling task, such advanced information is not required.
>
>
> > `I did not see a discussion on the societal impact in the paper, other than one part that the authors made disclaimer about the model's timeline alignment and this might not be the actual timeline. The author present a section in the supplementary file about this. The authors could expand on this, maybe describing that because the groundtruth timelines were also obtained by web crawling, they may not be precise enough, and a potential wrong prediction of news alignment may indeed have negative societal impact. For example, it might be used as evidence, while it is not indeed a confident reference.`
>
> Thank you for your suggestion. We update the corresponding section accordingly with a more in-depth discussion on potential societal impacts.

---

> ### Comment · Area_Chair_hNTc · 2023-08-29
>
> Dear Reviewer
>
> Kindly review and reply to the feedback provided by the authors.
>
> Regards
> AC

---

### Official Review · Reviewer_eQ6E · 2023-07-22
**a new timeline extraction task across multiple videos**

**Rating:** 7
**Confidence:** 5

**Strengths:**

1. Clever automation of data collection leading to a satisfyingly large dataset.
2. Convincing and comprehensive metrics.
3. Reasonable baseline performance.

**Additional Feedback:**

Please look at the order extraction literature to see if you need to cite it. Timeline extraction is a subset of the overall order extraction problem so there must be literature on comparison with ordered lists etc.

**Clarity:**

The paper is well written with some grammatical errors. The paper flows well and is easy to understand.

**Correctness:**

The claims made in the submission are correct. The dataset is constructed in a sound manner and the proposed metrics and methods are convincing.

**Documentation:**

In my view there is enough detail to ensure reproducibility.

**Ethics:**

I don't have any ethical concerns.

**Limitations:**

The authors acknowledge that the data is crawled from the web and hence is not completely under their control. I think that is healthy. It is beyond the scope of the paper to carefully curate each timeline to find out if there are objectionable opinions etc. The focus is on the timelines and there is no endorsement of the underlying content, which is as it should be.

The underlying problem of timeline extraction is content-neutral so my view is that the authors have adequately addressed the potential negative social impact.

**Opportunities For Improvement:**

1. The proposed baseline technique is adequate but not especially inspiring. Using a teacher student model is reasonable but surely that is not the only way to establish a baseline. These days you could apply a large multimodal model such as BLIP to establish a baseline as well. I don't see this as a serious weakness of the paper but am mentioning it for completeness.

2. There are grammatical errors such as:
Grammatical error on line 282
the nodes on a timeline is (should be are) ordered
Grammatical error on line 336
Perform fairly good (should be well)

They don't interfere with the reader's understanding of the paper but it would be nice to proofread carefully and eliminate such errors.

**Relation To Prior Work:**

The paper separates itself well from previous contributions. I would recommend that the authors look at prior ACM grand challenges to double check their novelty. I checked already and could not find anything that was at the scale and level of detail that the authors have proposed here.

**Summary And Contributions:**

The authors present a new timeline extraction task from multiple videos. They correctly point out that the timeline extraction task is not the same as video summarization. They make the following contributions:
1. Clever automatic extraction of a large dataset. That is important because it is the expense of manual collection that holds a lot of data collection back. The authors make clever use of the metadata to create curated timeline ground truth.
2. Four complementary metrics for assessment of success in timeline extraction. The fourth metric, pair-wise comparison truly rounds out the metrics.
3. A sound baseline method for extraction of the timeline based on a student teacher model.

---

> ### Author Response · Authors · 2023-08-23
>
> We genuinely thank you for recognizing our important contributions about data collection, evaluation protocol, and reasonable baselines.
>
> > `The proposed baseline technique is adequate but not especially inspiring. Using a teacher student model is reasonable but surely that is not the only way to establish a baseline. These days you could apply a large multimodal model such as BLIP to establish a baseline as well. I don't see this as a serious weakness of the paper but am mentioning it for completeness.`
>
> We appreciate the constructive feedback on our baseline choices. As the pioneering effort in formulating the video timeline modeling task and establishing the benchmark, our primary goal was to lay a solid foundation for future research in this direction. Hence, we developed baselines that are both straightforward and effective, ensuring an accessible starting point for the community. We recognize the potential of advanced multimodal models like BLIP and value your suggestion. As this research area evolves, we will certainly explore and integrate such approaches in the future.
>
>
>
> > `The paper separates itself well from previous contributions. I would recommend that the authors look at prior ACM grand challenges to double check their novelty. I checked already and could not find anything that was at the scale and level of detail that the authors have proposed here.`
>
> Thanks for bringing this up. We also checked and did not find tasks that largely overlap with our proposed task. By the way, we have added discussions with more relevant tasks, such as multi-video summarization, in Section 2 for a more comprehensive comparison with prior works.
>
> > `Please look at the order extraction literature to see if you need to cite it.`
>
> We did a search but did not find any relevant literature. Could you provide some specific pointers? We will add the discussion with them in our revision. Thank you.
>
> ----
>
> Thank you for pointing out several grammar errors. We have corrected them accordingly.

---

> ### Comment · Area_Chair_hNTc · 2023-08-29
>
> Dear Reviewer
>
> Kindly review and reply to the feedback provided by the authors.
>
> Regards
> AC

---

### Official Review · Reviewer_aTSU · 2023-07-25
**Video Timeline Modeling For News Story Understanding**

**Rating:** 10
**Confidence:** 3
**Correctness:** NA

**Strengths:**

The novelty of the paper is presenting the problem which has a great importance given the abundant of video from various social network that can capture different time point of a story.

**Additional Feedback:**

NA

**Clarity:**

I found the paper well written; explain the problem clearly as well as describe the three methods respectively.

**Documentation:**

section 3.2 describe data pipeline and how they have crawled, and created video and text sets and the algorithm in order to build the ground truth.

and the data is available https://github.com/google-research/google-research/tree/master/video_timeline_modeling however the code to create the data is not available.

**Limitations:**

The only limitation I see is about the time resolution; in the paper the time resolution has been set to 1 hour. The paper has not explore how the model can adapt by lowering this time length.

**Opportunities For Improvement:**

The paper does not discuss the sensitivity of the method at time spans. Their benchmark models are rather classifier with 24 classes. (number of nodes on the timelines). The question is the models are sensitive for smaller time lines. For example if models should be able to capture at 5 minutes interval ? Theoretically, such models should be able to map text and pieces of video from a 90 minutes soccer match.

**Relation To Prior Work:**

Yes; they have a dedicated section (2) where they go through the "Related Work"

**Summary And Contributions:**

The paper introduce a new challenge - which is to construct a video-associated timeline that represents the critical events and their evolutionary order in a news. They also provide a data along with three method in order to establish a benchmark for future work.

---

> ### Author Response · Authors · 2023-08-23
> **Clarification on the time resolution**
>
> Thank you for your positive feedback on the significance of our work.
>
> > `The paper does not discuss the sensitivity of the method at time spans. Their benchmark models are rather classifier with 24 classes. (number of nodes on the timelines). The question is the models are sensitive for smaller time lines. For example if models should be able to capture at 5 minutes interval? Theoretically, such models should be able to map text and pieces of video from a 90 minutes soccer match.`
>
> As pointed out by the reviewer, we formulated our proposed models to perform a 24-class classification task. This is because the maximum number of nodes on the timelines in our YouTube-News-Timeline dataset is 24. We did not explicitly set the time resolution. To be specific, the separation of nodes on a timeline is determined by the semantics of underlying events, instead of a fixed time interval.
>
> The reviewer's point on the potential requirement for finer temporal granularity, such as in the context of a 90-minute soccer match, is well-taken. Such a scenario would indeed necessitate a model capable of capturing events at, perhaps, 5-minute intervals or even shorter. While our current models are not directly optimized for such fine-grained timeline construction, we believe the foundational principles and methodologies we've introduced can be adapted for these purposes. By training on datasets curated for finer temporal granularity and adjusting our model architecture to predict a higher number of timeline nodes, we can potentially achieve the desired sensitivity for tasks like mapping text and video snippets from a soccer match or similar scenarios.

---

> ### Comment · Area_Chair_hNTc · 2023-08-29
>
> Dear Reviewer
>
> Kindly review and reply to the feedback provided by the authors.
>
> Regards
> AC

---

> > ### Comment · Reviewer_aTSU · 2023-08-29
> >
> > As authors clarified this is an important characteristics of the dataset that is limited to only 24 nodes; Perhaps as a caption to Figure 3 ? or elsewhere to emphasize on the limitation of data.This can also be a message for future work perhaps in the "outlook" section.
> >
> > Other than that I still believe, the paper introduce an interesting challenge that worth attention

---

> > > ### Author Response · Authors · 2023-08-29
> > > **Thanks**
> > >
> > > Thank you for the suggestion. We will further highlight this in our revision.

---

### Official Review · Reviewer_84mW · 2023-07-27
**Solid and well-motivated contribution, however more comprehensive dataset information is needed**

**Rating:** 7
**Confidence:** 4
**Clarity:** The paper is well written and clear.

**Strengths:**

- The paper addresses a novel and practical problem of video timeline modeling. The introduced YouTube-News-Timeline dataset can be a valuable resource for further research in this area.
- The paper experiments with three proposed deep learning approaches that seem well-motivated and designed to capture dependencies among videos and nodes on timelines.
- The paper is well-written and easy to follow. The presentation of the problem formulation, dataset creation, and methodology are well-described and clear.

**Additional Feedback:**

How does the long range of video duration (from 3 seconds to 12 hours) affect the model performance? Do the proposed baselines clip videos to a specific length, and what is the threshold used in terms of the number of frames (and/or sampling window)?

**Correctness:**

The claims appear correct, and the evaluation metrics seem reasonable for evaluating the performance of the proposed approaches. The explanations of the strategies used to ensure the automatic collection of the dataset is of high quality could be further improved with validation mechanisms put in place after each step, e.g., to detect any outliers or any steps of the dataset collection that might induce any biases.

**Documentation:**

I could not find a license that accompanies the dataset release. There is also no explanation provided for why the total amount of compute and the type of resources used are not detailed, can the authors comment on model capacity and training times?

**Limitations:**

The supplementary material contains a brief discussion of the potential ethical considerations, but more information on how the use of the proposed video timeline modeling could impact user privacy, data security, or the spread of misinformation would be helpful. News content can be subject to various forms of bias, and this can carry over into the modeling process. The authors should also address how they plan to mitigate potential biases in the video timeline modeling.

**Opportunities For Improvement:**

- While the paper contains a description of the YouTube-News-Timeline dataset, it would be helpful to include a more comprehensive summary of the main characteristics of the dataset, such as the diversity of the news/topics captured as well as finer-grained details such as whether the timelines include recent events (chronological information). There might also be concerns regarding potential biases in the dataset that have not been addressed.
-  The dataset's size seems rather small, and the diversity and representativeness could impact the generalizability of the proposed models. If the dataset is limited to specific news sources or regions, the models may not perform as effectively on videos from other sources or different contexts.
- Consider extending the related work section to provide a subsection with multi-video summarization methods and highlight the differences of the proposed timeline modeling task.

**Relation To Prior Work:**

The paper could include a more comprehensive discussion and comparison with related work in multi-video summarization, video classification, and video representation tasks.

**Summary And Contributions:**

This paper proposes a video timeline modeling task to organize and comprehend news videos. It introduces a benchmark dataset, YouTube-News-Timeline, proposes corresponding evaluation metrics, and presents three deep learning approaches, V-Transformer, Tri-Transformer, and Tri-Transformer + cross-modal distillation, to address the problem.

---

> ### Author Response · Authors · 2023-08-23
> **More dataset characteristics, more discussion with related work**
>
> We appreciate your recognition of the clarity of our paper and our contribution to the problem formulation, dataset, and methodologies.
>
> > `... helpful to include a more comprehensive summary of the main characteristics of the dataset...`
>
> As suggested, we added more dataset characteristics, including news sources ([Figure 8](https://ibb.co/sKKF26x)), topics ([Figure 9](https://ibb.co/VVqrLqs)), and chronological information about events ([Figure 10](https://ibb.co/s3rNwmz)) in the supplementary material. Importantly, our reference timelines are sourced from a diverse set of publishers, cover a wide array of topics, and span an expansive temporal range, ensuring diversity and alleviating inherent biases.
>
> We acknowledge the importance of addressing potential biases in the dataset, especially when dealing with news content. As shown above, our reference timelines are curated from various publishers and topics, ensuring diversity and minimizing inherent biases towards any particular source. While we've taken steps to diversify our references, we understand that biases in video content can still exist. To this end, we've appended a disclaimer to our paper, emphasizing our position. As pointed out by Reviewer eQ6E, “The authors acknowledge that the data is crawled from the web and hence is not completely under their control. I think that is healthy. It is beyond the scope of the paper to carefully curate each timeline to find out if there are objectionable opinions etc. The focus is on the timelines and there is no endorsement of the underlying content, which is as it should be.”
>
> > `The dataset's size seems rather small, and the diversity and representativeness...`
>
> As shown above, we've ensured diversity in our dataset by sourcing samples from a wide range of publishers. While the absolute size of the dataset might seem modest in comparison to some large-scale datasets in other domains, it's essential to highlight that the YouTube-News-Timeline dataset is the first of its kind in this specific area. With over 12k timelines and 300k videos, it provides a substantial and diverse foundation for the novel task of video timeline modeling.
>
> > `...related work in multi-video summarization...`
>
> Thanks for the constructive suggestion. We provide a brief comparison here. We have added a more comprehensive discussion of these research directions in Section 2.
> * General video classification and representation tasks focus on understanding individual videos. In contrast, the video timeline modeling task considers the intricate relationships among multiple videos.
> * While both multi-video summarization and video timeline modeling consider multiple videos as input, the former primarily aims to create a concise representation of multiple videos by selecting keyframes or short segments, emphasizing the overlap and complementarity among the videos. In comparison, video timeline modeling seeks to identify, order, and represent significant events in an evolutionary sequence.
>
> > `...potential ethical considerations...`
>
> Thank you for emphasizing the need for a more in-depth discussion on ethical considerations. We added more discussion in the supplementary and shared our idea on the possible plans.
>
>
>
> > `...license...can the authors comment on model capacity and training times?`
>
> The license is in the submission form. Based on the comment, we further clarify the license in the supplementary material. The YouTube-News-Timeline dataset is under the CC BY 4.0 International license.
>
> Our omission of the model capacity and training cost was not intentional. To make it clear, there are in total $2614784$ learnable parameters in our Tri-Transformer model. We trained such a model on 8 A100 GPUs for 3h to achieve convergence (100 epochs).
>
>
> > `How does the long range of video duration (from 3 seconds to 12 hours) affect the model performance? Do the proposed baselines clip videos to a specific length, and what is the threshold used in terms of the number of frames (and/or sampling window)?`
>
> We did not clip videos and applied a pre-trained video feature extractor to get video embeddings. Since it is an in-house method, due to potential intellectual property concerns, we are unable to share the details of the frame sampling rate. To assess the impact of video duration on our model's performance, we present a correlation between the video length and the corresponding video-level performance of the Tri-Transformer model, in terms of the Euclidean distance. As shown in the added [Figure 11](https://ibb.co/0cjB6bS) in the supplementary material, most videos are less than 20 minutes. The video-level Euclidean distance, based on the prediction of our Tri-Transformer model, is not sensitive to video length. We can observe that the variance becomes large for longer videos as the longer videos are less in both the training and test sets.
>
>
> ------------
> Thank you for your constructive comments.

---

> ### Comment · Area_Chair_hNTc · 2023-08-29
>
> Dear Reviewer
>
> Kindly review and reply to the feedback provided by the authors.
>
> Regards
> AC

---

### Author Response · Authors · 2023-08-23
**Revision submitted and response posted**

Dear Reviewers,

Thank you for your constructive feedback. We have provided detailed responses to each of your comments and have made corresponding revisions to our paper and supplementary material. All changes have been highlighted in red for clarity. We truly value your contributions to enhancing our work.

Best,

Authors

---

### Decision · Program_Chairs · 2023-09-22

**Decision:**

Accept (Spotlight)

**Comment:**

This paper receives good reviews from multiple reviewers and the authors have managed to address most of the concerns. The main contribution is a dataset to a new task that is of value. While there is concern of the limited application of the dataset (e.g., news recommendation), researchers from the community may discover more interesting applications this the availability of the new dataset. The authors are encouraged to include more details about the dataset in the final version.